# Exploring the impact of a life education program on the resilience of nursing students

**Yao-Mei Chuang[1], Wei-Hsiang Huang**[ORCID][2,3]*

**1** Department of Health and Leisure Management, Yuan-Pei University of Medical Technology, Hsinchu City, Taiwan, **2** Department of Early Childhood Care and Education, University of Kang-Ning, Taipei City, Taiwan, **3** Fire Department, New Taipei City Government, New Taipei City, Taiwan

* drddh@g.ukn.edu.tw

## Abstract

This study explores the impact of a life education program on nursing students' psychological resilience, focusing on meaning in life, life attitudes, and positive psychology. A quasi-experimental design with pre- and post-tests was used, involving 87 nursing students (40 in the experimental group, 47 in the control group) aged 20–25 years, with 85% being female. The experimental group attended a five-week life education program (500 minutes total). The Purpose in Life Test, Life Attitude Scale, and Positive Mental Health Scale were used as measurement tools, and data were analyzed using Generalized Estimating Equations (GEE). The results showed significant interaction effects in the experimental group on meaning in life (B = 4.09, p < 0.001), life attitudes (B = 11.29, p < 0.001), and positive psychology (B = 4.81, p = 0.009). Paired t-tests further confirmed significant within-group improvements, while the control group showed no changes. This study recommends integrating life education into nursing curricula and providing teacher training to enhance course delivery. Collaboration with mental health organizations is also suggested to offer additional support. Future research should expand the sample and explore qualitative methods to deepen understanding of life education's impact.

## Introduction

In recent years, life education has become an essential discipline in Taiwan, particularly in the field of nursing education. It not only explores the development across the life cycle but also emphasizes the quality of life and the understanding of death [1]. The content often integrates philosophical perspectives, including spiritual wisdom and Buddhist doctrines, aiming to help students develop life wisdom, practice life care, and acquire skills for end-of-life care [2]. In nursing education, life education has been recognized as a vital approach to equipping students with psychological resilience, which is essential for managing the emotional and ethical challenges of patient care, particularly in palliative and end-of-life settings [3].

**Data availability statement:** "The datasets generated and/or analyzed during this study are not publicly available because of the regulations stipulated by the Research Ethics Review Committee of the Hualien Tzu Chi Hospital Buddhist Tzu Chi Medical Foundation. Requests for data can be sent to the Research Ethics Review Committee of the Hualien Tzu Chi Hospital Buddhist Tzu Chi Medical Foundation via [irb@tzuchi.com.tw]. Data may be provided on reasonable request to eligible researchers and with the completion of all required prerequisites (https://hlm.tzuchi.com.tw/rec/index.php/regulations)."

**Funding:** The author(s) received no specific funding for this work.

**Competing interests:** The authors have declared that no competing interests exist.

A growing body of research highlights the importance of life education in shaping nursing students' attitudes toward life. A South Korean study found that life education and mental health significantly influence nursing students' perspectives on life, yet fewer than 30.4% of students have received life-related education. The study recommends systematically incorporating life education programs to help students cultivate a positive outlook on life [4]. Similarly, research in China emphasizes the need to integrate life education into nursing curricula, as it helps students find meaning in their work and supports their personal and professional development [5]. However, existing studies lack a clear framework for how life education programs directly enhance nursing students' psychological resilience, which is necessary for managing stress, death anxiety, and professional burnout [6]. To address this gap, the present study investigates the impact of a structured life education program on psychological resilience, specifically in the domains of meaning in life, attitudes toward life, and positive psychology.

**Psychological resilience**, a key concept in this study, refers to the ability to maintain a stable psychological state while facing adversity, particularly in end-of-life care situations. It encompasses three major components: meaning in life, attitudes toward life, and positive psychology [7]. Studies indicate that resilience is closely linked to a strong sense of meaning in life, a positive outlook, and the application of positive psychological strategies, all of which contribute to personal growth and self-actualization [7], Targeted educational interventions, such as life education programs, have been shown to strengthen resilience, reduce compassion fatigue, and improve overall well-being among nursing students [8].

**Meaning in life** is a fundamental component of psychological resilience and is associated with lower levels of depression, hopelessness, and suicidal ideation [9]. In nursing education, fostering meaning in life is particularly important, as students frequently encounter death and must develop effective coping strategies to provide compassionate end-of-life care [10]. Research suggests that interventions incorporating reflective practices, narrative storytelling, and experiential learning can enhance students' sense of meaning in life and improve their emotional resilience [11,12].

**Attitudes toward life**, another critical dimension of psychological resilience, influence how individuals perceive and respond to life and death [4,13]. Studies suggest that life education can reduce death anxiety, improve life satisfaction, and help nursing students develop a more accepting and positive attitude toward mortality [14,15]. Cultivating a positive life attitude enables nursing students to manage emotional distress effectively, enhancing their coping mechanisms and job satisfaction [16].

Positive psychology, the third key concept explored in this study, focuses on well-being and emotional resilience. Research has shown that fostering positive psychology through life education programs can enhance students' ability to handle stress and improve their mental health [17]. Structured interventions that incorporate mindfulness, self-reflection, and peer discussions have been found to strengthen resilience and promote professional readiness among nursing students [18–20].

Despite the growing recognition of life education's importance, few empirical studies have evaluated its direct impact on nursing students' psychological resilience in Taiwan. Most existing research remains theoretical, with limited investigation into how life education influences key dimensions of resilience, such as coping strategies and professional identity formation [21]. This study aims to bridge this gap by assessing the effectiveness of a life education program in enhancing psychological resilience among nursing students, focusing on meaning in life, attitudes toward life, and positive psychology. The findings of this study have the potential to inform curriculum development and guide educators in integrating life education into nursing programs.

The significance of this study lies in its contributions to both theory and practice. First, it provides empirical evidence on the effectiveness of structured life education programs in enhancing psychological resilience among nursing students, offering insights into curriculum design. Second, these findings inform educational institutions on best practices for integrating life education into nursing curricula, ensuring that students develop the essential skills needed to navigate the emotional and ethical challenges of their profession.

## Materials and methods

### Research design

This study employed a quasi-experimental design with pre- and post-tests for both experimental and control groups. The research methodology adopted an experimental research approach, where the independent variable was the intervention of the life education program, and the dependent variable was psychological resilience, encompassing meaning in life, attitude toward life, and positive psychology. The data for these variables, except for the categorical variables representing the background characteristics of the participants, were all continuous variables.

Participants in the experimental group were those who chose to enroll in the elective life education course and thus received the intervention. The control group, composed of students who did not enroll in the course, did not participate in any related activities during the intervention period. Moreover, because the control group did not attend class sessions during the intervention, this minimized potential interference or cross-contamination between the groups. This distinction allowed the study to better isolate the effects of the life education program, ensuring that the control group remained unaffected by any components of the intervention.

### Participants and eligibility criteria

Given the regional setting and convenience for the researcher, the study was conducted at the university of technology where the researcher is employed. Participants were selected using a convenience sampling method. However, convenience sampling may introduce potential bias as students who volunteer to participate may have characteristics that differ from those who do not, thus limiting the generalizability of the findings. Therefore, the results of this study are applicable primarily to students from similar backgrounds and under similar conditions.

Inclusion criteria were as follows: (a) university students enrolled in the 2018 academic year who had not previously participated in any life education program; (b) clear consciousness and the ability to communicate in Mandarin or Taiwanese; and (c) signed informed consent. Participants who did not sign the consent form were excluded from the study.

### Sample size estimation

The sample size for this study was determined using G*Power 3.1 to ensure adequate statistical power for detecting intervention effects. Based on previous literature, intervention studies targeting university students have typically involved sample sizes ranging from 35 to 82 participants [22–24]. To ensure that our study met rigorous statistical criteria, sample size estimation was performed for both between-group and within-group comparisons.

For between-group comparisons, an independent sample t-test was used to assess differences between the experimental and control groups at pre-test and post-test. The required sample size was calculated assuming a medium-to-large

effect size (Cohen's d = 0.65), a power of 0.80, and an alpha level of 0.05. The results indicated that a minimum total sample size of 78 participants was necessary, with 36 assigned to the experimental group and 42 to the control group, reflecting an allocation ratio of 1.175.

For within-group comparisons, a paired sample t-test was used to evaluate changes within each group over time. Using the same statistical parameters (Cohen's dz = 0.65, power = 0.80, α = 0.05), the G*Power analysis determined that at least 21 participants per group were required. Given these calculations and considering potential attrition, a total of 97 students were initially recruited, and 87 participants completed the study, resulting in a retention rate of 89.69%.

The final sample size exceeded the minimum requirement for both statistical tests, ensuring a statistical power above 0.80 to detect meaningful intervention effects. Additionally, while the final sample size falls within the range reported in previous studies, future research may benefit from a larger sample to enhance the robustness and generalizability of the findings.

### Intervention

The life education program designed for this study was developed with a structured approach, drawing from the concept of meaning in life, and the core principles of life education, including its origins and foundational values [25]. The course content was informed by three key perspectives emphasized in previous literature: social, philosophical, and psychological [26]. Additionally, the program incorporated the researcher's practical experience in nursing education, resulting in the development of five main courses. These covered essential topics such as attitudes toward life and death, the life cycle, rights of the dying and issues related to suicide, coping with loss, and alternative views on immortality and life perspectives.

The intervention was deliberately structured based on well-established frameworks from the literature on life education. By following a clear curriculum and avoiding personal interpretation or ad-hoc teaching methods, the program aimed to minimize subjective influence and ensure consistency in delivery.

Previous research offers various suggestions for the duration of life education interventions in nursing. Some studies recommend a once-a-week format over five consecutive weeks [27], while others advocate for two-hour sessions weekly [28]. In this study, a 10-session format was adopted, consisting of two 50-minute sessions per week for five weeks (a total of 500 minutes). A detailed breakdown of the course design is provided in Table 1.

### Research tools

Three validated and reliable scales were used in this study, all of which have been culturally adapted to fit the local context.

The first tool was the **Purpose in Life Test (PIL)**, originally developed by foreign scholars [29]. This scale was translated into Chinese by domestic researchers [30], and has been widely used in the local academic context with demonstrated validity and reliability. The PIL includes five dimensions: zest for life, life goals, autonomy, avoidance, and future expectations. In this study, the Cronbach's α was 0.87, indicating good internal consistency. Higher scores indicate a stronger sense of meaning in life. The PIL uses a 7-point scale with 20 items, resulting in a total score range of 20–140.

The second tool was the **Life Attitude Scale (LAS)**, also based on a foreign scale [31], and translated into a concise Chinese version by domestic scholars [32]. The LAS has been used in multiple local studies, showing its applicability in the cultural context. It covers six dimensions: ideal, autonomy, love and care, existence, attitude toward death, and life experiences. The Cronbach's α in this study ranged from 0.68 to 0.80, with strong construct validity (GFI = 0.93, CFI = 0.93, RMSEA = 0.053), confirming the reliability of the scale. Higher scores indicate a more positive attitude toward life. The LAS uses a 5-point scale with 24 items, with a total score range of 24–120.

The third tool, the **Positive Mental Health Scale**, was originally developed by domestic scholars [33] and is specifically designed for the local context. This scale includes five dimensions: self-acceptance, interpersonal relationships, emotional

**Table 1. Summary of the life education course for nursing students.**

| Course | Objectives | Teaching content related to the intervention | Intervention variable |
|---|---|---|---|
| Attitudes toward life and death | 1. Modern Perspectives on Life<br>2. Attitudes Toward Death and Their Influencing Factors<br>3. Exploring and Clarifying Personal Perspectives on Life and Attitudes Toward Death<br>4. Understanding Individual Attitudes Toward Death and Life and Their Interrelationship | 1. Pre-class Reading: The Last 14 Tuesday Sessions<br>2. The Origin of Life<br>3. Changes in Attitudes Toward Life and Death from Agricultural to Industrial Societies<br>4. The Impact of Modern Medicine on Contemporary Attitudes Toward Life and Death | Life attitudes |
| The cycle of life | 1. The Value of Personal Life<br>2. Personal Meaning of Life<br>3. The Reasons for Continuing the Family Line | 1. How does modern society view and address the issue of being childless differently?<br>2. If you are unable to conceive but wish to raise a child, what options would you choose?<br>3. What are your views on surrogacy and assisted reproductive technology? Do you support them? What are the legal, social, ethical, and commercial concerns?<br>4. What are the reasons for choosing not to have children, even if one is capable of conceiving?<br>5. Assisted Reproduction and Abortion<br>6. What is your perspective on Frankl's statement, "Life itself has meaning, and it is not enhanced by having children; if an individual's life lacks meaning, having children will not bring personal meaning"? Why? | Meaning in life |
| Rights of the dying and suicide issues | 1. Diverse Perspectives on the Right to Death<br>2. Strengthening Resilience Against Suicide | 1. Pre-class Reading: (1) Visible Darkness - A Journey Through Depression (2) Night Falls Suddenly; Understanding Suicide - Analysis and Explanation<br>2. Group Discussions: Explore different aspects of suicide from the perspectives of psychology, religion, socio-culture, ethics, and an integrated approach.<br>3. Group Presentations<br>4. Synthesize Group Reports and Provide Further Explanation on Suicide Prevention to Enhance Students' Ability to Prevent Suicide. | Meaning in life |
| loss in life | 1. Enhancing Individual Acceptance of Various Types of Loss<br>2. Strengthening Individual Coping Abilities After Experiencing Loss | 1. Meditation Activity: Reflecting on the Growth Process and Past Experiences of Loss<br>2. Writing Exercise: Document the Moments of Loss Experienced During Growth<br>3. Group Activity: Individual Sharing - "What I Have Lost Is…"<br>4. Group Sharing<br>5. The Story of Dharma Master Cheng Yen | Life attitudes |
| Alternative immortality and life | 1. Perspectives on Organ Donation and the Ability for Diverse Thinking<br>2. Clarifying the Value of Life<br>3. Integrating Personal Life Philosophy | 1. Understanding Organ Donation Cards: Learn what an organ donation card is and whether you have one. Explain the significance and scope of organ donation, and provide a brief overview of the organ donation regulations.<br>2. Class Discussion: Explore the legal, religious, cultural, and ethical aspects of organ donation. | Positive psychology |

balance, family harmony, and optimism. It has been validated with strong psychometric properties, including GFI = 0.91, AGFI = 0.90, SRMR = 0.05, CFI = 0.96, and RMSEA = 0.07. In this study, the Cronbach's α ranged from 0.76 to 0.92, indicating excellent internal consistency. The Positive Mental Health Scale uses a 5-point scale with 25 items, resulting in a total score range of 25–125. Given that this scale was designed within the local cultural context, it inherently meets the cultural adaptation requirements. Higher scores reflect better mental and physical health.

 

## Statistical analysis

Data analysis was performed using IBM SPSS for Windows version 22.0. The reliability of the research instruments was assessed through Cronbach's α coefficient, which evaluated internal consistency, and test-retest reliability, which examined instrument stability over time. Descriptive statistics, including frequency distributions, percentages, means, and standard deviations, were used to summarize participant characteristics and provide an overview of the dataset.

For group comparisons, independent sample t-tests were conducted to evaluate differences between the experimental and control groups at pre-test and post-test. Paired sample t-tests were performed to assess within-group changes over time, measuring the intervention's effectiveness.

To account for within-subject correlations in repeated measures data, Generalized Estimating Equations (GEE) were applied. GEE was chosen over repeated measures ANOVA and mixed-effects models due to its ability to handle correlated observations, accommodate missing data, and provide robust standard error estimates when normality assumptions are violated [34]. The working correlation matrix was set to an exchangeable structure to model within-subject dependencies, ensuring a stable estimation of intervention effects.

To control for baseline differences between groups, pre-test values of psychological resilience variables were included as covariates in the model. Baseline homogeneity between groups was assessed before the analysis, and any significant differences were adjusted for in the final model. The GEE model included time (pre-test and post-test), group (experimental and control), and their interaction as fixed factors to determine whether the life education program produced statistically significant improvements in meaning in life, attitudes toward life, and positive psychology. Statistical significance was set at $p < 0.05$.

## Research ethics

This study received ethical approval on March 5, 2019, from the Research Ethics Review Committee of the Hualien Tzu Chi Hospital Buddhist Tzu Chi Medical Foundation (approval number: IRB108–26-B). Conducted between March 5, 2019, and December 31, 2019, the research involved the administration of questionnaires as part of the intervention. Written informed consent was obtained from all participants, who were all adult college students. No minors participated in this study.

Participants were thoroughly informed about the study's objectives, their rights, and were encouraged to respond to the assessment tools with honesty and independence. To reduce social desirability bias, all questionnaires were anonymized, and no personal identifiers such as names or student IDs were collected. This anonymity ensured that participants could respond freely without concern for how their answers would be perceived. Additionally, the use of validated and reliable questionnaires further minimized potential bias, supporting the validity and integrity of the data collected.

Participants were assured of their right to withdraw from the study at any time without any negative consequences. All collected data was securely stored, accessible only to the research team, with stringent privacy protections in place. This study adhered strictly to the ethical guidelines and regulations of the host country, ensuring full compliance with the highest ethical standards throughout the research process.

## Results

### Demographic distribution

The personal demographic data of the participants in the experimental and control groups, including gender, personal religious beliefs, family religious beliefs, birth order, family economic status, and experiences with the death of relatives or friends, are presented in Table 2. The analysis results indicate that the experimental and control groups are homogeneous in terms of gender, personal religious beliefs, family religious beliefs, birth order, family economic status, and experiences with the death of relatives or friends. Chi-square tests and Fisher's exact tests showed no significant differences between

Table 2. Summary of distribution and differences of the research objects in terms of basic personal information (N = 87).

| Basic personal information/category | All participants (*N* = 87) | Group | | χ² | p-value |
| --- | --- | --- | --- | --- | --- |
| | | Experimental group (n = 40) | Control group (n = 47) | | |
| **Gender** | | | | 0.001 | 0.970 |
| male | 11 (12.64) | 5 (12.50) | 6(12.77) | | |
| female | 76 (87.36) | 35 (87.50) | 41(87.23) | | |
| **Individual Religious Beliefs** | | | | 1.460 | 0.480 |
| Folk beliefs in Taiwan (including others) | 32(36.78) | 17(42.50) | 15(31.91) | | |
| Catholicism/Christianity | 17(19.54) | 6(15.00) | 11(23.40) | | |
| Taoism/Buddhism | 38(43.68) | 17(42.50) | 21(44.68) | | |
| **Family Religious Beliefs** [a] | | | | | |
| Folk beliefs in Taiwan (including others) | 24(27.59) | 12(30.00) | 12(25.53) | | |
| Catholicism/Christianity | 13(14.94) | 4(10.00) | 9(19.15) | | |
| Taoism/Buddhism | 50(57.47) | 24(60.00) | 26(55.32) | | |
| **Birth Order in the Family** | | | | 0.140 | 0.930 |
| First (including only child) | 44(50.57) | 21(52.50) | 23(48.94) | | |
| Second | 22(25.29) | 10(25.00) | 12(25.53) | | |
| Third (including later) | 21(24.14) | 9(22.50) | 12(25.53) | | |
| **Family Economic Status** | | | | 1.460 | 0.230 |
| Above average | 69(79.31) | 34(85.00) | 35(74.47) | | |
| Disadvantaged | 18(20.69) | 6(15.00) | 12(25.53) | | |
| **Experience of Relatives' or Friends' Death** | | | | 3.120 | 0.080 |
| Yes | 72(28.76) | 30(75.00) | 42(89.36) | | |
| No | 15(17.24) | 10(25.00) | 5(10.64) | | |

Note: Chi-square tests (χ²) were used for categorical variables when the expected cell count in all cells was ≥ 5; otherwise, Fisher's exact test was applied.

these variables across the two groups (p > 0.05), suggesting that the participants in both groups were similar in these background characteristics, thus ruling out potential confounding effects on the experimental outcomes.

## Reliability analysis of research tools

After the formal implementation of the study, reliability analyses were conducted on the Meaning in Life Scale, Life Attitude Scale, and Positive Mental Health Scale.

For the Meaning in Life Scale, Cronbach's α values were 0.60 (pre-test) and 0.59 (post-test) for the experimental group, and 0.52 (pre-test) and 0.58 (post-test) for the control group. Despite these values being below the commonly accepted threshold of 0.70, the test-retest reliability (γ = 0.91 for the experimental group and γ = 0.76 for the control group) shows strong stability over time.

The Life Attitude Scale and Positive Mental Health Scale demonstrated better internal consistency, with Cronbach's α ranging from 0.88 to 0.98 across both groups, and test-retest reliability (γ) of 0.92 for the Life Attitude Scale and 0.87 for the Positive Mental Health Scale, confirming their robustness.

It's important to note that Cronbach's α is influenced by the number of items in a scale [35]. The Meaning in Life Scale has 20 items, which may explain its lower α compared to the 24-item Life Attitude Scale and 25-item Positive Mental Health Scale. Nevertheless, the high test-retest reliability indicates the Meaning in Life Scale's temporal consistency.

In summary, while the Meaning in Life Scale's Cronbach's α values are slightly below the accepted threshold, the strong test-retest reliability confirms its stability and suitability for measuring psychological resilience over time.

## Changes over time for the participants

To understand the changes over time in the experimental and control groups following the life education program intervention, independent sample t-tests were used to analyze changes between groups, and paired sample t-tests were used to analyze changes within groups, as shown in Table 3.

The analysis of group differences in pre-test scores revealed no significant differences between the experimental and control groups in meaning in life (t = -0.17, p = 0.860), life attitude (t = -1.15, p = 0.250), and positive psychology (t = 0.74, p = 0.460), indicating homogeneity in pre-test scores between the groups. However, in the post-test scores, meaning in life (t = 2.26, p = 0.030) and positive psychology (t = 3.10, p = 0.010) showed significant differences, indicating that the intervention had a significant effect on these variables.

Within-group analysis of the experimental group revealed significant changes in meaning in life (t = 5.40, p < 0.001), life attitude (t = 5.59, p < 0.001), and positive psychology (t = 4.68, p < 0.001). In contrast, the control group showed no significant changes, indicating that the intervention led to more significant changes in the experimental group compared to the control group.

## Effectiveness of the Intervention

To assess the intervention's effectiveness on psychological resilience (including meaning in life, life attitude, and positive psychology), generalized estimating equations (GEE) were applied using a compound symmetry working correlation matrix to account for time effects and robust standard errors for accurate significance testing [36], as shown in Table 4. The key measure of effectiveness was the interaction between group (experimental vs. control) and time (pre- and post-test).

The results indicated no significant group differences in the pre-test for meaning in life (B = -0.31, p = 0.886), life attitude (B = -6.41, p = 0.243), or positive psychology (B = 1.97, p = 0.443), confirming baseline comparability. No significant time effects were found in the control group for meaning in life (B = -0.64, p = 0.539), life attitude (B = 0.49, p = 0.805), or positive psychology (B = -0.96, p = 0.563).

**Table 3. Changes of the two groups at different time points (N = 87).**

| Variable | Experimental group (n = 40) | Control group (n = 47) | Between-group t-value | p |
|---|---|---|---|---|
| | M ± SD | M ± SD | | |
| Meaning in life | | | | |
| Pre-test | 86.18 ± 9.89 | 86.49 ± 7.19 | -0.17 | 0.860 |
| Post-test | 89.63 ± 8.71 | 85.85 ± 6.87 | 2.26* | 0.030 |
| Within-group t-value (p) | 5.40 (<0.001) | -0.61 (0.550) | | |
| Life attitudes | | | | |
| Pre-test | 126.55 ± 30.21 | 132.96 ± 19.50 | -1.15 | 0.250 |
| Post-test | 138.32 ± 21.39 | 133.45 ± 21.92 | 1.05 | 0.300 |
| Within-group t-value (p) | 5.59 (<0.001) | 0.24 (0.810) | | |
| Positive psychology | | | | |
| Pre-test | 81.50 ± 10.53 | 89.53 ± 13.63 | 0.74 | 0.460 |
| Post-test | 95.35 ± 8.70 | 88.57 ± 11.67 | 3.10** | 0.010 |
| Within-group t-value (p) | 4.68 (<0.001) | -0.57 (0.570) | | |

Note: Independent sample t-tests were used for between-group comparisons, and paired sample t-tests were used for within-group comparisons.

**Table 4. Comparison of the magnitude of changes of measurement indicators at various times.**

| Variable | Estimated parameter B (95%CI) | Standard error (SE.) | Wald χ² | p-value |
|---|---|---|---|---|
| Meaning in life | | | | |
| Intercept term | 86.49(84.46 to 88.52) | 1.04 | 6849.54 | <0.001 |
| Group: Experimental Group vs. Control Group | -0.31(-3.96 to 3.33) | 1.86 | 0.03 | 0.866 |
| Time: Post-test vs. Pre-test | -0.64(-2.68 to 1.40) | 1.04 | 0.38 | 0.539 |
| Group×Time: Experimental Group×Post-test vs. Control Group×Post-test | 4.09(1.71 to 6.47) | 1.22 | 11.31 | <0.001 |
| Life attitudes | | | | |
| Intercept term | 132.96(127.44 to 138.47) | 2.81 | 2233.23 | <0.001 |
| Group: Experimental Group vs. Control Group | -6.41(-17.17 to 4.36) | 5.49 | 1.36 | 0.243 |
| Time: Post-test vs. Pre-test | 0.49(-3.40 to 4.38) | 1.98 | 0.06 | 0.805 |
| Group×Time: Experimental Group×Post-test vs. Control Group×Post-test | 11.29(5.65 to 16.92) | 2.87 | 15.43 | <0.001 |
| Positive psychology | | | | |
| Intercept term | 89.53(85.68 to 93.39) | 1.97 | 2072.07 | <0.001 |
| Group: Experimental Group vs. Control Group | 1.97(-3.06 to 6.99) | 2.56 | 0.59 | 0.443 |
| Time: Post-test vs. Pre-test | -0.96(-4.20 to 2.29) | 1.66 | 0.33 | 0.563 |
| Group×Time: Experimental Group×Post-test vs. Control Group×Post-test | 4.81(1.19 to 8.42) | 1.84 | 6.79 | 0.009 |

Note: Generalized Estimating Equations (GEE) were used to analyze repeated measures data with an exchangeable working correlation structure. Baseline values of psychological resilience variables were included as covariates. Robust standard errors were applied to obtain reliable estimates. The table presents regression coefficients (B), standard errors (SE), 95% confidence intervals (CI), and p-values. Statistical significance: $p < 0.05$, $p < 0.01$.

Significant interaction effects were observed for meaning in life (B = 4.09, p < 0.001), life attitude (B = 11.29, p < 0.001), and positive psychology (B = 4.81, p = 0.009), indicating the intervention's positive impact.

It's important to note that the different B values reflect the distinct score ranges of the scales: the Meaning in Life Scale (20–140), Life Attitude Scale (24–120), and Positive Mental Health Scale (25–125). Larger score ranges, such as for meaning in life, tend to produce smaller B values. Therefore, the magnitude of B should be interpreted in the context of the scale's structure. Despite the variation in B values, the significant interaction effects show that the intervention was effective across all three outcomes.

## Discussion

### Enhancement of meaning in life through life education

This study demonstrates that the life education intervention significantly enhances the meaning in life, life attitudes, and positive psychology among nursing students, aligning with previous research findings. The findings support existing literature indicating that life education deepens students' understanding of spirituality and resilience, contributing to both mental health and professional development in healthcare contexts [3]. Consistent with previous studies, life education programs have shown both immediate and long-term benefits, particularly in enhancing students' coping abilities in end-of-life care settings [6]. Since meaning in life is considered a key determinant of resilience when facing death-related challenges [37], this intervention underscores the importance of helping nursing students develop a strong sense of professional identity and emotional preparedness [21].

### Improvement in attitudes toward death and coping mechanisms

The study also found that life education improves attitudes toward death, which is a critical factor influencing healthcare professionals' decision-making and emotional responses in palliative care settings [15]. Research suggests that life education effectively mitigates death anxiety and fosters a more accepting attitude toward mortality [16]. Moreover, structured

discussions on death-related topics have been emphasized as a crucial component of palliative care training, helping students process their fears and gain deeper insights into their professional roles [14]. Therefore, these findings reinforce the importance of integrating life education into nursing curricula to enhance students' psychological resilience and professional competence.

### Strengthening positive psychology and emotional resilience

The results also indicate that life education significantly enhances positive psychology, a key factor in mental well-being and adaptability in clinical settings. Positive psychology interventions emphasize emotional intelligence and resilience, both of which are crucial for nursing students facing emotionally challenging situations [8]. Previous studies confirm that life education reduces negative emotions, fosters death acceptance, and enhances self-efficacy in end-of-life care [12]. These findings support the inclusion of positive psychology principles in life education programs to help nursing students develop stronger coping mechanisms for handling workplace stress and patient loss.

### Effects of course structure and teaching strategies

This study's intervention design further contributes to existing literature by demonstrating the effects of an adjusted life education program structure. Previous research found that a five-week program (one 150-minute session per week) improved university students' life satisfaction but had limited impact on life attitudes [20]. In this study, a revised intervention format (two 100-minute sessions per week for five weeks, totaling 10 sessions) led to significant improvements across multiple psychological domains. These results suggest that increased session frequency and interactive teaching methods, such as the Death Café approach, group discussions, and reflective exercises, facilitate deeper engagement and learning [37]. The curriculum covered essential topics including death education, end-of-life care, and creative therapies (e.g., psychodrama, photo voice), which further enhanced students' self-efficacy and interest in palliative care careers [12].

### Implications for nursing education

In summary, this study confirms the significant benefits of life education in improving nursing students' meaning in life, attitudes toward death, and positive psychology. These results highlight the importance of embedding life education into nursing curricula, providing students with essential skills to manage the emotional and ethical challenges of their profession. Additionally, adjusting course frequency and incorporating interactive teaching strategies further strengthened the intervention's impact, reinforcing the value of structured life education programs in nursing education.

## Conclusion and suggestions

### I. Conclusion

The life education program intervention demonstrated significant improvements in nursing students' meaning in life, life attitudes, and positive psychology. These results underscore the importance of integrating life education into nursing curricula, not only to enhance students' holistic development but also to prepare them for the emotional and professional demands of palliative care. By fostering greater resilience and emotional regulation, this intervention lays a strong foundation for nursing students as they transition into their careers, particularly in end-of-life care contexts.

### II. Recommendations and limitations

#### (I) Practical recommendations.

1. For Educational Institutions: It is recommended that life education be included as a required or elective course in higher education institutions, particularly for nursing students. Providing professional training for educators will improve the

effectiveness of course delivery and ensure that students receive comprehensive education in life care and psychological resilience.

2. For Universities and Colleges: Counseling and support systems should be integrated into life education courses to address students' personal and emotional challenges. Collaboration with mental health organizations and community groups can provide additional resources and support, further enhancing the students' understanding of life education and its practical applications.

**(II) Research recommendations and limitations.**

1. **Generalizability:** The study's findings are limited by the specific regional and institutional context in which participants were drawn, using convenience sampling. This method may introduce selection bias, making the results less representative of broader populations. Future research should extend to diverse educational and geographical contexts, and adopt randomized or stratified random sampling methods to improve generalizability.

2. **Sample Size and Validity**: Although the sample size met the statistical requirements for a medium effect size, it is relatively small for a quantitative study. This may reduce the robustness of the findings and increase the risk of Type II errors. Future studies should increase the sample size and extend the intervention period to improve the statistical power, representativeness, and reliability of the results.

3. **Methodological Considerations and Social Desirability Bias:** The reliance on self-report measures presents a risk of social desirability bias, even though anonymous questionnaires were used. Additionally, the quantitative approach may not fully capture the emotional and cognitive changes experienced by students during life education. To address this, future research should combine qualitative methods, such as interviews or observations, which would not only reduce social desirability bias but also provide a deeper, more nuanced understanding of the intervention's effects.

4. **Confounding Variables**: Although the study controlled for primary variables, certain confounding factors, such as participants' prior exposure to death or personal coping mechanisms, were not accounted for. Future studies should include more detailed baseline assessments to better control for these potential confounders and provide a clearer picture of the intervention's effectiveness.

5. **Cultural and Contextual Factors**: While the tools used were adapted from international scales and locally validated, the content and methods of life education may still be influenced by specific cultural and institutional contexts. Future studies should further examine how life education programs can be adapted and applied across different cultural and social environments to ensure wider applicability.

6. **Long-Term Effectiveness**: This study assessed outcomes immediately after the intervention, leaving questions about the long-term sustainability of the observed benefits. Future research should adopt longitudinal designs to determine whether improvements in meaning in life, life attitudes, and positive psychology persist over time. Extending the intervention period beyond five weeks may also allow students more time to internalize the material, potentially leading to more lasting changes in psychological resilience.

7. **Sampling and Potential Bias:** The dual role of the researcher as both the program developer and implementer may have unintentionally influenced program delivery, and convenience sampling may have introduced selection bias. In future research, involving independent facilitators and employing more robust sampling methods, such as randomized or stratified random sampling, would enhance objectivity and strengthen the reliability of the findings.

Despite these limitations, this study provides valuable insights into the impact of life education programs and lays the foundation for future research to further explore and enhance life education's role in nursing education.

## Acknowledgments

We would like to extend our heartfelt gratitude to Professor Hu Yih-Jin and Assistant Professor Tseng Chie-Chien from the Department of Health Promotion and Health Education at National Taiwan Normal University for their invaluable expertise and guidance in shaping this paper. We also wish to thank the nursing students at the University of Technology who participated in this study for their contributions.

## Author contributions

**Conceptualization:** Yao-Mei Chuang.

**Data curation:** Yao-Mei Chuang.

**Formal analysis:** Wei-Hsiang Huang.

**Funding acquisition:** Yao-Mei Chuang.

**Investigation:** Yao-Mei Chuang.

**Methodology:** Yao-Mei Chuang.

**Project administration:** Yao-Mei Chuang.

**Resources:** Yao-Mei Chuang.

**Validation:** Wei-Hsiang Huang.

**Visualization:** Wei-Hsiang Huang.

**Writing – original draft:** Yao-Mei Chuang.

**Writing – review & editing:** Wei-Hsiang Huang.

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
