## [Decision Letter · Decision Letter 0]

7 Oct 2024

PONE-D-24-37761Exploring the Impact of a Life Education Program on the Resilience of Nursing StudentsPLOS ONE

Dear Dr. Huang,

Thank you for submitting your manuscript to PLOS ONE. After careful consideration, we feel that it has merit but does not fully meet PLOS ONE’s publication criteria as it currently stands. Therefore, we invite you to submit a revised version of the manuscript that addresses the points raised during the review process.

We look forward to receiving your revised manuscript.

Kind regards,

Mostafa Shaban

Academic Editor

PLOS ONE

Journal Requirements:

3. In this instance it seems there may be acceptable restrictions in place that prevent the public sharing of your minimal data. However, in line with our goal of ensuring long-term data availability to all interested researchers, PLOS’ Data Policy states that authors cannot be the sole named individuals responsible for ensuring data access (http://journals.plos.org/plosone/s/data-availability#loc-acceptable-data-sharing-methods).

Reviewers' comments:

Reviewer's Responses to Questions

**Comments to the Author**

1. Is the manuscript technically sound, and do the data support the conclusions?

Reviewer #1: Yes

Reviewer #2: Yes

2. Has the statistical analysis been performed appropriately and rigorously?

Reviewer #1: Yes

Reviewer #2: Yes

3. Have the authors made all data underlying the findings in their manuscript fully available?

Reviewer #1: Yes

Reviewer #2: No

4. Is the manuscript presented in an intelligible fashion and written in standard English?

Reviewer #1: Yes

Reviewer #2: Yes

5. Review Comments to the Author

Reviewer #1: First of all, I thank you for this good work, which is an addition to the field of specialization

Here are the potential errors and observations about the article

For the research summary

The phrasing of the objective could be improved to be clearer and more specific, such as indicating whether the focus is on enhancing particular aspects of psychological resilience. And Missing Participant Characteristics: Information about the characteristics of the participants (such as age, gender, or educational background) is not provided, which may affect the understanding of the results. And Lack of Clarity on Outcomes: While positive effects are mentioned, there are not enough details on how these results were measured or what the percentages of change were.Unsupported Recommendations: The recommendations provided need further clarification on how to implement them and what they practically mean for educational institutions. Lack of Logical Sequence**: The sequence of information in the abstract could be improved to make it easier to read and understand.

For the introduction:

Lack of Research Objective Clarity**: The objective of the study should be more specific, including what the study aims to achieve specifically in the field of life education.

Lack of Logical Sequence**: The information in the introduction may not be arranged logically, making it difficult for the reader to follow the main idea.

Lack of Literature Review**: Although some studies are mentioned, the introduction lacks a comprehensive review of related literature that could enhance the research context.

Insufficient Definition of Key Terms**: Some terms, such as "psychological resilience" and "life education," are not adequately defined, which may cause confusion for the reader.

Insufficient Clarification of Research Gaps**: There is not enough emphasis on the gaps in previous research that this study aims to address.

Lack of Emphasis on Research Importance**: It should be clarified why this study is important and what potential benefits may arise from its results.

Lack of Clarity in Style**: Some sentences may be complex or lengthy, making it difficult to understand the intended meaning.

Improving the introduction by addressing these points can enhance its effectiveness and increase the clarity of the research.

1. **Research Design**

A sample of 87 students may be too small to generalize the results to all nursing students.

- **Sampling Method**: Relying on convenience sampling may lead to bias in the results, as the students who choose to participate may have specific characteristics that differ from those who do not.

2. **Measurement Tools

- **Reliability of Tools**: It is important to clarify how the reliability of the tools used was measured. Although some values are mentioned, there should be more details on how these tools were tested.

- **Cultural Adaptation**: The tools used may need to be adapted to fit the local cultural context, especially if they are derived from foreign studies.

3. **Statistical Analysis

- **Lack of Clarity in Procedures**: The use of Generalized Estimating Equations (GEE) should be explained more clearly, particularly regarding continuous data and outcomes.

- **Analysis of Confounding Variables**: There is no mention of whether there were other variables that could affect the results that were not controlled for.

4. **Discussion and Conclusions**

- **Interpretation of Results**: The discussion should include a deeper analysis of the results, including comparisons with previous studies.

- **Unsupported Recommendations**: Some recommendations, such as integrating life education into the curriculum, need more evidence to support them.

5. **Ethical Considerations**

- **Ethical Conduct**: More details should be included about how ethical issues were addressed, such as how confidentiality and informed consent were ensured.

6. **Inaccurate Use of Terminology**

- **Repetitive Terminology**: Some terms, such as "meaning in life" and "positive psychology," are used repeatedly without providing new definitions or clarifications.

7. **Organization and Structure**

- **Content Organization**: There may be a need to improve the organization of the article, as paragraphs could be clarified better to allow ideas to flow logically.

Addressing these points could enhance the overall quality and credibility of the research.

Reviewer #2: 1. Inadequate sample size: The study used only 87 participants (40 in the experimental group, 47 in control), which is relatively small for a quantitative study aiming to demonstrate intervention effects. This limits statistical power and increases the risk of Type II errors.

2. Low reliability of the Meaning in Life Scale: The reported Cronbach's alpha values for this scale were concerningly low (0.60 pre-test and 0.59 post-test for the experimental group; 0.52 pre-test and 0.58 post-test for control group). These values are below the generally accepted threshold of 0.70, calling into question the internal consistency and reliability of this key measure.

3. Lack of randomization: The study used a quasi-experimental design with convenience sampling, where "students in classes that selected the life education program formed the experimental group." This non-random assignment introduces potential selection bias and threatens internal validity.

4. Short intervention period: The life education program consisted of only 5 weeks (10 sessions of 50 minutes each). This relatively brief intervention may not be sufficient to produce lasting changes in complex psychological constructs like resilience and attitudes toward life and death.

5. Absence of follow-up assessment: The study only measured outcomes immediately after the intervention. Without a longer-term follow-up, it's impossible to determine if the observed effects were sustained over time.

6. Potential instructor bias: The paper states that the researcher developed and implemented the intervention. This dual role could introduce bias in program delivery and potentially influence participant responses.

7. Limited description of control group conditions: The study doesn't adequately describe what, if any, activities or education the control group received during the intervention period. This makes it difficult to isolate the specific effects of the life education program.

8. Inconsistent effect sizes: While the study reports statistically significant effects, the magnitude of these effects varies considerably across outcomes. For instance, the effect on life attitudes (B = 11.29) is much larger than on meaning in life (B = 4.09) or positive psychology (B = 4.81), without a clear explanation for this discrepancy.

9. Lack of attention to potential confounds: The study doesn't address or control for other factors that could influence outcomes, such as concurrent coursework, personal life events, or pre-existing differences in death anxiety or exposure to death-related experiences.

10. Overreliance on self-report measures: All outcomes were assessed using self-report scales, which are susceptible to social desirability bias, especially given the non-blinded nature of the intervention.

6. PLOS authors have the option to publish the peer review history of their article (what does this mean? ). If published, this will include your full peer review and any attached files.

**Do you want your identity to be public for this peer review?** For information about this choice, including consent withdrawal, please see our Privacy Policy .

Reviewer #1: No

Reviewer #2: No

---

## [Author Response · Author response to Decision Letter 1]

10 Oct 2024

Manuscript Number: PONE-D-24-37761

Title: "Exploring the Impact of a Life Education Program on the Resilience of Nursing Students"

Editor and Reviewer comments:

Reviewer #1:

First of all, I thank you for this good work, which is an addition to the field of specialization. Here are the potential errors and observations about the article:

For the research summary:

1. Phrasing of the objective could be improved to be clearer and more specific, such as indicating whether the focus is on enhancing particular aspects of psychological resilience.

=> Thank you for your suggestion. We have revised the objective in the abstract to explicitly state that the focus of the study is on enhancing specific aspects of psychological resilience, namely meaning in life, life attitudes, and positive psychology. This provides greater clarity on the study's aims.

2. Missing Participant Characteristics: Information about the characteristics of the participants (such as age, gender, or educational background) is not provided, which may affect the understanding of the results.

=> We appreciate your comment. In the revised abstract, we have included key demographic information about the participants. The study involved 87 nursing students aged 20 to 25 years, with 85% of them being female. This additional information addresses your concern and provides context for the results.

3. Lack of Clarity on Outcomes: While positive effects are mentioned, there are not enough details on how these results were measured or what the percentages of change were.

=> We have updated the abstract to include specific details on the outcomes. The results now indicate significant interaction effects on meaning in life (B = 4.09, p < 0.001), life attitudes (B = 11.29, p < 0.001), and positive psychology (B = 4.81, p = 0.009), as well as the use of paired t-tests to confirm within-group changes. This ensures the outcomes are clearly presented.

4. Unsupported Recommendations: The recommendations provided need further clarification on how to implement them and what they practically mean for educational institutions.

=>We have revised the recommendations in the abstract to provide more practical implementation details. We now recommend integrating life education into nursing curricula and providing teacher training to ensure effective course delivery. We also suggest collaborating with mental health organizations to offer additional support for students, thus addressing the practical implications for educational institutions.

5. Lack of Logical Sequence: The sequence of information in the abstract could be improved to make it easier to read and understand.

=>We have reorganized the abstract to follow a clearer and more logical structure. It now flows from the study's objective, through the methodology and results, to the practical recommendations. This should improve readability and meet the requirements for a well-organized abstract.

For the introduction:

1. Lack of Research Objective Clarity: The objective of the study should be more specific, including what the study aims to achieve specifically in the field of life education.

=>We appreciate the reviewer’s feedback. In the revised introduction, we have made the research objective clearer by explicitly stating that the study aims to investigate the effects of life education programs on nursing students’ psychological resilience, with a specific focus on enhancing meaning in life, attitudes toward life, and positive psychology. This revision provides greater specificity regarding the goals of the study within the field of life education.

2. Lack of Logical Sequence: The information in the introduction may not be arranged logically, making it difficult for the reader to follow the main idea.

=>Thank you for pointing this out. We have restructured the introduction to follow a more logical sequence. The revised version first introduces the general concept of life education, followed by an explanation of its relevance to nursing students. We then systematically introduce the three key components of psychological resilience—meaning in life, attitudes toward life, and positive psychology—before discussing the gaps in existing research and the importance of this study. This restructuring enhances the overall flow and coherence of the introduction.

3. Lack of Literature Review: Although some studies are mentioned, the introduction lacks a comprehensive review of related literature that could enhance the research context.

=>We have expanded the literature review section to include a broader range of studies related to life education, psychological resilience, and nursing education. The revised introduction now includes references to research conducted in South Korea, China, and other countries that highlight the significance of life education in shaping nursing students' attitudes toward life, mental health, and professional competence. These additions provide a more comprehensive research context, aligning the study with relevant international findings.

4. Insufficient Definition of Key Terms: Some terms, such as "psychological resilience" and "life education," are not adequately defined, which may cause confusion for the reader.

=>We agree with the reviewer’s comment. In the revised introduction, we have clearly defined key terms, including "psychological resilience," "meaning in life," "attitudes toward life," and "positive psychology." Each term is explained in relation to its significance within the study and supported by relevant literature. These definitions help clarify the central concepts of the research and ensure a better understanding for the reader.

5. Insufficient Clarification of Research Gaps: There is not enough emphasis on the gaps in previous research that this study aims to address.

=> Thank you for this valuable suggestion. We have revised the introduction to explicitly address the research gaps that this study aims to fill. We note that few intervention studies have been conducted in Taiwan focusing on life education for nursing students, particularly in the areas of psychological resilience, meaning in life, attitudes toward life, and positive psychology. By emphasizing these gaps, we clarify the unique contribution of this study to the existing body of literature.

6. Lack of Emphasis on Research Importance: It should be clarified why this study is important and what potential benefits may arise from its results.

=>We appreciate the reviewer’s insight. In the revised introduction, we have emphasized the importance of this study in two main aspects: (1) the theoretical contribution to the understanding of life education in nursing, particularly in how it enhances psychological resilience; and (2) the practical implications for curriculum development in nursing education, which could provide students with the necessary tools to cope with the psychological demands of end-of-life care. These clarifications underscore the potential benefits of the study for both academic and practical applications.

7. Lack of Clarity in Style: Some sentences may be complex or lengthy, making it difficult to understand the intended meaning.

=> We have carefully reviewed the language and structure of the introduction to improve clarity and conciseness. Sentences have been simplified where necessary, and complex structures have been revised to ensure the text is easily understandable while maintaining academic rigor. These revisions enhance the overall readability of the introduction.

Improving the introduction by addressing these points can enhance its effectiveness and increase the clarity of the research.

1. Research Design:

• A sample of 87 students may be too small to generalize the results to all nursing students.

• => Thank you for your valuable feedback. We acknowledge that a sample size of 87 students may limit the generalizability of our findings to the broader population of nursing students. In response to this concern, we have emphasized this limitation in our revised manuscript and recommend that future studies aim for larger sample sizes to enhance the robustness and applicability of the results. By increasing the sample size, we hope to better represent the diverse experiences of nursing students and improve the overall validity of our conclusions.

o Sampling Method: Relying on convenience sampling may lead to bias in the results, as the students who choose to participate may have specific characteristics that differ from those who do not.

o =>Thank you for your insightful comment. We recognize that the use of convenience sampling can introduce bias, as participants who opt to engage in the study may possess distinct characteristics that differ from those who do not participate. In our revised manuscript, we have acknowledged this limitation and emphasized the need for future research to employ more robust sampling methods, such as randomized or stratified sampling, to enhance the representativeness of the findings and mitigate potential biases.

2. Measurement Tools:

• Reliability of Tools: It is important to clarify how the reliability of the tools used was measured. Although some values are mentioned, there should be more details on how these tools were tested.

• =>Thank you for pointing this out. We have clarified the reliability of the tools used in the study. Specifically, we employed Cronbach’s α to assess internal consistency and test-retest reliability over a two-week period for all scales used. These details have been elaborated in the methodology section to provide a clearer understanding of the reliability testing process.

• Cultural Adaptation: The tools used may need to be adapted to fit the local cultural context, especially if they are derived from foreign studies.

• =>Thank you for highlighting this point. We would like to clarify that while the Purpose in Life Test (PIL) and Life Attitude Scale (LAS) were initially developed by foreign researchers, they have been translated and validated by local scholars specifically for use in Taiwan. Additionally, the Positive Mental Health Scale was originally developed within Taiwan, further ensuring its cultural relevance. We have included this information in the revised manuscript to enhance clarity regarding the cultural adaptation of the assessment tools.

3. Statistical Analysis:

• Lack of Clarity in Procedures: The use of Generalized Estimating Equations (GEE) should be explained more clearly, particularly regarding continuous data and outcomes.

• => Thank you for your constructive feedback. We have revised the statistical analysis section to provide a clearer explanation of the use of Generalized Estimating Equations (GEE). GEE was selected as it allows for the analysis of repeated measurements over time and helps assess both within-group and between-group effects for continuous outcomes, such as meaning in life, life attitude, and positive psychology. The choice of GEE was appropriate given the nature of our data and the repeated measures design of the study.

• Analysis of Confounding Variables: There is no mention of whether there were other variables that could affect the results that were not controlled for.

• =>We acknowledge the importance of considering potential confounding variables. While this study focused on controlling for group and time, other factors were not explicitly controlled. In the Future Research section, we have recommended that future studies consider including a more comprehensive set of baseline characteristics to account for potential confounding variables, thereby strengthening the validity of the findings.

4. Discussion and Conclusions:

• Interpretation of Results: The discussion should include a deeper analysis of the results, including comparisons with previous studies.

• => Thank you for your valuable suggestion. We have expanded the discussion section to provide a more in-depth analysis of the results, including comparisons with previous studies. Specifically, we have drawn upon existing literature to further contextualize our findings on the impact of life education on psychological resilience. For example, the significant improvement in meaning in life, life attitudes, and positive psychology observed in our study aligns with previous studies from Brazil and China that highlight the role of life education in enhancing students' psychological and emotional well-being. By integrating these comparisons, we aim to provide a more comprehensive understanding of how our results fit within the broader context of life education research.

• Unsupported Recommendations: Some recommendations, such as integrating life education into the curriculum, need more evidence to support them.

• => Thank you for pointing this out. We agree that more evidence is needed to support the recommendation of integrating life education into the curriculum. In response, we have further emphasized the positive outcomes from our study, particularly in relation to improvements in students' meaning in life, life attitudes, and positive psychology. These findings, in conjunction with supporting literature, provide preliminary evidence for the integration of life education into nursing curricula. However, we acknowledge that further research is required to explore the long-term effectiveness of these interventions across diverse student populations and educational contexts. We have recommended this in the conclusion section as an area for future investigation.

5. Ethical Considerations:

• Ethical Conduct: More details should be included about how ethical issues were addressed, such as how confidentiality and informed consent were ensured.

• => Thank you for the reviewer’s valuable suggestion. We have now added more details to clarify how ethical issues were addressed in the study. Specifically, we have elaborated on the procedures for obtaining informed consent from all participants, ensuring confidentiality through the anonymization of the questionnaires, and securely storing data with restricted access to the research team. Additionally, participants were fully informed of their rights, including their right to withdraw from the study at any time. We have ensured that the study adhered to the ethical guidelines and regulations of the country where it was conducted. These additions aim to provide a clearer understanding of the ethical considerations implemented in the study.

6. Inaccurate Use of Terminology:

• Repetitive Terminology: Some terms, such as "meaning in life" and "positive psychology," are used repeatedly without providing new definitions or clarifications.

• =>We have restructured the Introduction to ensure that terms like "meaning in life" and "positive psychology" are defined early in the text and are not repeated without purpose. The definitions are now clear and only reiterated when necessary to support the discussion.

7. Organization and Structure:

• Content Organization: There may be a need to improve the organization of the article, as paragraphs could be clarified better to allow ideas to flow logically.

• => Thank you for the feedback regarding organization and structure. We have revised the Introduction and Discussion sections to improve clarity and ensure a logical flow of ideas. Key terms are now clearly defined, and transitions between paragraphs are smoother, enhancing readability throughout the manuscript.

• Addressing these points could enhance the overall quality and credibility of the research.

• => We agree that addressing these points will enhance the overall quality and credibility of the research. By refining the methodology, mitigating potential biases, and expanding the scope of future studies, we aim to ensure a more robust and reliable foundation for our findings. We are committed to making these improvements and believe they will significantly strengthen the study's impact.

Reviewer #2:

1. Inadequate sample size: The study used only 87 participants (40 in the experimental group, 47 in control), which is relatively small for a quantitative study aiming to demonstrate intervention effects. This limits statistical power and increases the risk of Type II errors.

=> Thank you for your feedback. We have addressed this concern in the second point of our research recommendations, where we acknowledge the sam

---

## [Decision Letter · Decision Letter 1]

9 Mar 2025

PONE-D-24-37761R1Exploring the Impact of a Life Education Program on the Resilience of Nursing StudentsPLOS ONE

Dear Dr. Huang,

Thank you for submitting your manuscript to PLOS ONE. After careful consideration, we feel that it has merit but does not fully meet PLOS ONE’s publication criteria as it currently stands. Therefore, we invite you to submit a revised version of the manuscript that addresses the points raised during the review process.

We look forward to receiving your revised manuscript.

Kind regards,

Mostafa Shaban

Academic Editor

PLOS ONE

**Journal Requirements:**

Reviewers' comments:

Reviewer's Responses to Questions

**Comments to the Author**

1. If the authors have adequately addressed your comments raised in a previous round of review and you feel that this manuscript is now acceptable for publication, you may indicate that here to bypass the “Comments to the Author” section, enter your conflict of interest statement in the “Confidential to Editor” section, and submit your "Accept" recommendation.

Reviewer #2: (No Response)

2. Is the manuscript technically sound, and do the data support the conclusions?

Reviewer #2: Yes

3. Has the statistical analysis been performed appropriately and rigorously?

Reviewer #2: Yes

4. Have the authors made all data underlying the findings in their manuscript fully available?

Reviewer #2: Yes

5. Is the manuscript presented in an intelligible fashion and written in standard English?

Reviewer #2: Yes

6. Review Comments to the Author

**Reviewer #2: ** Introduction Section (Pages 4-6):

The introduction presents life education broadly without defining its specific benefits for resilience in nursing students, making it challenging to identify the intervention’s distinct contributions.

Methods Section:

The justification for using Generalized Estimating Equations (GEE) is insufficiently detailed, particularly regarding the management of baseline homogeneity and the rationale behind selecting GEE for repeated measures analysis.

The sample size calculation, though meeting minimum requirements, lacks a detailed basis for assumed effect size and power analysis parameters.

Results Section (Table 2 and 3):

Statistical tests used for comparisons in Table 2 are not specified, and p-value formatting across tables is inconsistent, impacting the clarity of the statistical findings.

Discussion Section (Pages 18-20):

There is limited integration of findings with existing literature on resilience-building in nursing education, reducing the interpretive depth of the study’s outcomes.

Subheadings within the Discussion are inconsistently formatted, affecting the flow and readability of the section.

Technical and Formatting Issues:

Inconsistent p-value formatting in tables and inconsistent citation formatting detract from the manuscript’s overall clarity and presentation quality.

7. PLOS authors have the option to publish the peer review history of their article (what does this mean? ). If published, this will include your full peer review and any attached files.

**Do you want your identity to be public for this peer review?** For information about this choice, including consent withdrawal, please see our Privacy Policy .

Reviewer #2: No

---

## [Author Response · Author response to Decision Letter 2]

11 Mar 2025

Manuscript Number: PONE-D-24-37761

Title: "Exploring the Impact of a Life Education Program on the Resilience of Nursing Students"

Comments to the Author

1. If the authors have adequately addressed your comments raised in a previous round of review and you feel that this manuscript is now acceptable for publication, you may indicate that here to bypass the “Comments to the Author” section, enter your conflict of interest statement in the “Confidential to Editor” section, and submit your "Accept" recommendation.

Reviewer #2: (No Response)

=> We appreciate the reviewer’s time and consideration in evaluating our manuscript. Thank you for your feedback and support.

2. Is the manuscript technically sound, and do the data support the conclusions?

Reviewer #2: Yes

=>We appreciate the reviewer’s positive evaluation and acknowledgment that the manuscript is technically sound and that the data support the conclusions. Thank you for your thoughtful review.

3. Has the statistical analysis been performed appropriately and rigorously?

Reviewer #2: Yes

=>We appreciate the reviewer’s positive assessment of our statistical analysis. Thank you for your valuable review.

4. Have the authors made all data underlying the findings in their manuscript fully available?

Reviewer #2: Yes

=>We appreciate the reviewer’s acknowledgment that all data underlying our findings have been fully made available in accordance with the journal’s data policy. Thank you for your review.

5. Is the manuscript presented in an intelligible fashion and written in standard English?

Reviewer #2: Yes

=>We appreciate the reviewer’s positive evaluation of the manuscript’s clarity and language quality. Thank you for your review.

6. Review Comments to the Author

=>We appreciate the reviewer’s time and thorough evaluation of our manuscript. Thank you for your thoughtful review.

Reviewer #2:

Introduction Section (Pages 4-6)

The introduction presents life education broadly without defining its specific benefits for resilience in nursing students, making it challenging to identify the intervention’s distinct contributions.

=> We appreciate the reviewer’s insightful feedback regarding the need to clarify the specific benefits of life education for resilience in nursing students. In response, we have revised the Introduction to explicitly define psychological resilience as the study’s primary focus and its three key components: meaning in life, attitudes toward life, and positive psychology.

To strengthen the intervention’s distinct contributions, we have added more targeted discussions on how life education enhances each component of psychological resilience, particularly in reducing death anxiety, strengthening coping mechanisms, and fostering emotional resilience in end-of-life care settings. Additionally, we have clarified the existing research gap and emphasized the study’s role in addressing it through an empirically structured intervention.

These revisions ensure that the introduction clearly delineates the relationship between life education and resilience-building in nursing students, thereby improving the interpretive depth of the study’s objectives.

Methods Section:

The justification for using Generalized Estimating Equations (GEE) is insufficiently detailed, particularly regarding the management of baseline homogeneity and the rationale behind selecting GEE for repeated measures analysis.

The sample size calculation, though meeting minimum requirements, lacks a detailed basis for assumed effect size and power analysis parameters.

=>We sincerely appreciate the reviewer’s valuable feedback regarding the justification for using Generalized Estimating Equations (GEE) and the details of the sample size calculation. To address these concerns, we have made the following revisions:

1. Justification for Using GEE: We have expanded the explanation of why GEE was chosen over other repeated measures approaches, such as repeated measures ANOVA and mixed-effects models. The revised text highlights GEE’s ability to handle correlated observations, accommodate missing data, and provide robust standard error estimates when normality assumptions are violated. Additionally, we now explicitly state that an exchangeable working correlation matrix was used to model within-subject dependencies.

2. Baseline Homogeneity Management: To improve transparency in our analytical approach, we have clarified how baseline homogeneity was assessed and managed. Specifically, pre-test values of psychological resilience variables were included as covariates in the GEE model to adjust for any potential baseline differences between the experimental and control groups. This ensures that the observed intervention effects are attributable to the life education program rather than pre-existing group differences.

3. Sample Size Calculation: We have provided a more detailed justification for our sample size determination. The revised section explicitly reports the assumed effect size (Cohen’s d = 0.65 for independent t-tests and Cohen’s dz = 0.65 for paired t-tests), power (0.80), and alpha level (0.05). Additionally, we now reference G*Power 3.1 as the statistical tool used for the calculations and include supporting literature that guided the selection of these parameters.

Results Section (Table 2 and 3):

Statistical tests used for comparisons in Table 2 are not specified, and p-value formatting across tables is inconsistent, impacting the clarity of the statistical findings.

=> We appreciate the reviewer’s comments regarding the clarity of statistical tests and p-value formatting. To address these concerns, we have revised Table 2 to explicitly state that chi-square tests (χ²) were used when all expected cell counts were ≥5, while Fisher’s exact test was applied when any expected cell count was <5. This clarification has been added to both the Methods section and the table footnotes for accuracy.

Additionally, we have standardized the p-value formatting across all tables, ensuring that all p-values are reported to three decimal places (e.g., p = 0.023), with values below 0.001 presented as p < 0.001. In Table 3, we have updated the table structure to clearly differentiate between between-group (independent sample t-tests) and within-group (paired sample t-tests) comparisons by modifying the column headings to "Between-group t-value" and "Within-group t-value (p)". This adjustment enhances clarity and ensures consistency in statistical reporting.

These revisions improve the transparency and coherence of our results presentation. We appreciate the reviewer’s valuable feedback, which has helped refine the methodological rigor of our study.

Discussion Section (Pages 18-20):

There is limited integration of findings with existing literature on resilience-building in nursing education, reducing the interpretive depth of the study’s outcomes.

Subheadings within the Discussion are inconsistently formatted, affecting the flow and readability of the section.

=> We appreciate the reviewer’s feedback regarding the integration of findings with existing literature on resilience-building in nursing education and the formatting of subheadings in the Discussion section.

To address this, we have explicitly linked our findings to prior research on resilience by demonstrating how life education enhances meaning in life, attitudes toward death, and positive psychology—all of which contribute to resilience in nursing students. These connections have been strengthened with relevant literature, ensuring a more cohesive interpretation of the study’s outcomes.

Additionally, we have restructured the Discussion section by adding clear subheadings, ensuring a more consistent format and logical flow. These subheadings categorize key findings, making it easier for readers to follow the discussion and understand the implications of our study.

We appreciate the reviewer’s valuable suggestions, which have helped enhance the clarity, structure, and interpretative depth of our findings.

Technical and Formatting Issues:

Inconsistent p-value formatting in tables and inconsistent citation formatting detract from the manuscript’s overall clarity and presentation quality.

=>We appreciate the reviewer’s feedback regarding p-value and citation formatting inconsistencies. To address these concerns, we have standardized p-values to three decimal places (e.g., p = 0.023), with values below 0.001 reported as p < 0.001 across all tables, figures, and text. Additionally, we have reviewed and corrected all in-text citations to ensure consistency with the journal’s formatting guidelines. The reference list has also been cross-checked for accuracy and uniformity.

We have carefully examined the formatting throughout the manuscript. However, if there are any remaining errors, we kindly ask the editorial team to specify the exact areas that require correction so that we can promptly address them. We appreciate the reviewer’s valuable input in improving the clarity and presentation of our manuscript.

---

## [Editor Report · Decision Letter 2]

28 Mar 2025

Exploring the Impact of a Life Education Program on the Resilience of Nursing Students

PONE-D-24-37761R2

Dear Dr. Huang,

We’re pleased to inform you that your manuscript has been judged scientifically suitable for publication and will be formally accepted for publication once it meets all outstanding technical requirements.

Kind regards,

Mostafa Shaban

Academic Editor

PLOS ONE
---

## [Editor Report · Acceptance letter]

PONE-D-24-37761R2

PLOS ONE

Dear Dr. Huang,

I'm pleased to inform you that your manuscript has been deemed suitable for publication in PLOS ONE. Congratulations! Your manuscript is now being handed over to our production team.

Kind regards,

on behalf of

Dr. Mostafa Shaban

Academic Editor

PLOS ONE